# Exploration of the Compressive Strength and Microscopic Properties of Portland Cement Taking Attapulgite and Montmorillonite Clay as an Additive

**DOI:** 10.3390/ma16051794

**Published:** 2023-02-22

**Authors:** Jiahao Yan, Mengya Zhou, Jinyuan Fan, Ping Duan, Zuhua Zhang

**Affiliations:** 1Faculty of Materials Science and Chemistry, China University of Geosciences, Wuhan 430074, China; 2Key Laboratory of Advanced Building Materials of Anhui Province, Anhui Jianzhu University, Hefei 230022, China; 3Guangxi Key Laboratory of New Energy and Building Energy Saving, Guilin University of Technology, Guilin 541004, China; 4Key Laboratory of Advanced Civil Engineering Materials of Ministry of Education, School of Materials Science and Engineering, Tongji University, Shanghai 201804, China

**Keywords:** calcination, attapulgite, montmorillonite, cement, microstructure property, hydration

## Abstract

The effects of attapulgite and montmorillonite calcinated at 750 °C for 2 h as supplementary cementing materials (SCMs) on the working properties, mechanical strength, phase composition, morphology, hydration and heat release of ordinary Portland cement (OPC) were studied. The results show that pozzolanic activity increased with time after calcination, and with the increase in content of calcined attapulgite and calcined montmorillonite, the fluidity of cement paste exhibited a downward trend. Meanwhile, the calcined attapulgite had a greater effect on the decrease in the fluidity of cement paste than calcined montmorillonite, and the maximum reduction was 63.3%. Within 28 days, the compressive strength of cement paste with calcined attapulgite and montmorillonite was higher than that of the blank group in the later stage, and the optimum dosages of calcined attapulgite and montmorillonite were 6% and 8%, respectively. In addition, the compressive strength of these samples reached 85 MPa 28 days later. The introduction of calcined attapulgite and montmorillonite increased the polymerization degree of silico-oxygen tetrahedra in C-S-H gels during cement hydration, thereby contributing to accelerating the early hydration process. In addition, the hydration peak of the samples mixed with calcined attapulgite and montmorillonite was advanced, and the peak value was lower than that of the control group.

## 1. Introduction

Cement is known as the most widely used building material worldwide because of its good bonding ability. However, in the cement production process, the chemical conversion of limestone into lime leads to the emission of CO_2_ as a by-product [1], the chemical considered the main cause of the global greenhouse effect [2]. Studies have shown that producing one ton of cement clinker emits 900 kg of CO_2_, accounting for 7–8% of total global carbon emissions [3]. Therefore, in recent years, more and more materials with adhesive and potential pozzolanic activity have been used as substitutes for cement in the construction industry. Considering the wide range of sources, low price, abundant minerals, and environmental protection properties of natural mineral raw materials, they are extensively used to reduce the extensive energy consumption of the cement industry, and are considered capable of replacing cement as supplementary cementing materials (SCMs) and reducing the amount of cement and carbon emissions [4,5].

As a kind of natural mineral clay, the main chemical compositions of attapulgite and montmorillonite are (Mg,Al)_5_[(OH)_2_(Si,Al)_8_O_20_]·8H_2_O and (Na,Ca)_0.33_(Al,Mg)_2_[SiO_4_O_10_](OH)_2_·nH_2_O [6]. In terms of structure, the two share a high-level similarity: both of them are composed of two continuously arranged silicon–oxygen tetrahedrons that are sandwiched by an irregularly arranged layer of aluminum–oxygen octahedrons to form a 2:1 “sandwich” structure [7,8,9]. This interlayer structure is relatively fluffy, and the two have a similar structure, so they share similar performance as well. Some studies have reported that they can be used as SCMs to reduce the use of cement clinker. For example, the Zghair [10] team and others have shown that high-reactivity attapulgite (HRA) combined with increasing curing temperature can improve the compressive strength of cement slurry at later stages; ShihoKawashima et al. [11]. also studied the effect of adding a small amount of high-purity attapulgite on the adhesive properties of cement slurry and found that clay increased the cohesiveness and viscosity of cement slurry by adding attapulgite; Safi et al. [12]. used montmorillonite instead of cement clinker and found that adding montmorillonite reduced the compressive strength of these samples but greatly improved its water absorption, permeability and acid erosion. Meanwhile, the strengthening effect of montmorillonite was evaluated by means of dynamic mechanical analysis (DMA) by Yu et al. [13]. The compressive strength of the cementitious material was increased, but its compressive strength decreased with water solubility. In addition, according to the existing researches, both of them have water absorption [14,15] and expansion properties [13,16], and will therefore reduce the fluidity of cement slurry when used as SCMs, thereby affecting their large-scale application.

However, Safa et al. [17] found that calcined attapulgite clay filled the insides of cement mortar, improved surface micro-cracks, and enhanced the mechanical properties of cement mortar; Qin et al. [18] also found that the addition of calcined clay improved the fluidity of cementitious materials, especially the addition of calcined montmorillonite, which changed the microscopic morphology of hydration products such as C-S-H gels and C-A-H gels in the system while promoting the formation of Si-O bonds into Si-O-Si bonds, thus accelerating the early hydration process and improving the development of mechanical strength. Rimvydas [19] and Rodrigo et al. [20] also found that due to the existence of adsorbed water, pozzolanic activity was significantly enhanced with the application of calcined montmorillonite. Additionally, pore water, coordination water and structural water on both the inner and outer surfaces of attapulgite and montmorillonite were gradually removed as the calcination temperature increased, thus affecting the mineral structure. In this case, calcination can be applied to the improvement of the hydration activity of mineral clay and the reduction of the influence of a considerable clay content on the fluidity of cement slurry, which has been previously mentioned.

In this case, there is evidence that it is feasible to use calcination for the treatment of mineral raw materials to improve the performance of cement slurry. Moreover, attapulgite and montmorillonite feature a wide range of sources, low cost, abundant minerals and environmental protection properties and can also reduce the extensive energy consumption of the cement industry, thereby promoting the sustainable development of mineral clay, which is endowed with huge practical significance and great application potential. To this end, on the basis of a large number of experiments, attapulgite and montmorillonite were hereby placed in a Muffle oven and calcined at 750 °C for 2 h. The influence of the calcination mechanism of clay mineral materials on the hydration process of cement was revealed by preparing two calcined clay-cement-based binary composite cementitious materials. The calcined products were added to ordinary Portland cement with different dosages (mass fraction), and the effects of different dosages of calcined products (calcined attapulgite and montmorillonite) on the hydration, micro-properties and hydration heat release of ordinary Portland cement were explored.

## 2. Materials and Methods

### 2.1. Raw Materials

Herein, ordinary Portland cement (OPC) produced by Huaxin Cement Co, LTD with a relative density of 3100 kg/m^3^ was used. The basic mechanical properties of the cement are shown in Table 1. The hereby-adopted attapulgite mineral clay powder was produced by Dingbang Company in Changzhou, Jiangsu Province, and the montmorillonite mineral clay powder was produced by Kangtai Mineral Products Processing Plant in Lingshou County, Shijiangzhuang, Hebei Province. The main chemical components of OPC, attapulgite mineral clay and montmorillonite mineral clay are plotted in Table 2.

### 2.2. Calcination

The two clay powders were spread evenly on a porcelain plate. In order to ensure adequate calcination, the proportion of the powder placed on the porcelain plate should not exceed two-thirds. Additionally, complete dihydroxylation of mineral clays requires high calcination temperatures, but too-high calcination temperatures can cause particle sintering [21]. Therefore, a porcelain plate containing clay powder was placed in a Muffle oven at the room temperature of 20 °C for 146 min, calcined at 750 °C for 2 h, and then cooled to room temperature.

### 2.3. Mix Proportion and Preparation of the Samples

Calcined or uncalcined clay powder was mixed into the cement according to the mix ratio in Table 3 below, and this was then mixed in the cement paste mixer. Then, the paste was poured into a mold of 40 mm × 40 mm × 40 mm for further shaking and solidifying, and the mold was removed after being covered with plastic wrap for 24 h of curing to make a sample of binomic cemellating material. AG/MT was used to represent the cement paste sample made of uncalcined attapulgite and montmorillonite, CAG/CMT was used for the cement paste sample made of calcined clay material, and the following numbers were adopted to indicate its dosage (mass fraction). The ratio of water to binder materials was set as 0.4, and the sample preparation process is plotted in Figure 1.

### 2.4. Testing Method

#### 2.4.1. Determination of Pozzolanic Activity after Calcination

Herein, the residual quantity method summarized by Zhong [22] was adopted to determine the pozzolanic activity of auxiliary SCMs. The principle was that active substances in calcined clay would react in calcium hydroxide solution, and the obtained reacting products and alkali metals could be fully dissolved in hydrochloric acid solution, while the remaining parts would not react with hydrochloric acid. Through a series of dissolution and drying processes, the active component content of the sample was obtained, and its activity rate was derived. The derived formula is presented in Equation (1):(1)$$k_{\alpha }=\frac{\left(M_{A}-M_{C}\right)-\left(M_{A}\times k_{s}\right)}{M_{A}\times k_{H}}$$

Note: $k_{\alpha }$ denotes the pozzolanic activity measured using the residual measurement method; $M_{A}$, the total mass of the calcined clay powder sample; $M_{C}$, the mass of residual residue after filtration; $k_{s}$, the mass fraction of other soluble components in the calcined clay powder; and $k_{H}$, the mass fraction of the total SiO_2_ and Al_2_O_3_ in the calcined clay powder.

#### 2.4.2. Determination of the Fluidity of Cementitious Materials

The fluidity of the mixtures was determined in accordance with Chinese National Standard GB/T 2419-2005 by measuring the flow diameter.

#### 2.4.3. Mechanical Property Test

According to GB/T 17671-1999 “Cement Mortar Strength Test Method (ISO Method)”, the compressive performance of the test block was tested, and the samples prepared were placed in the center of the universal testing machine during the test. The compression test was conducted using a YAW4605 pressure testing machine with a running speed of 2.4 kN/s, and the average value of the three samples was taken as the test result.

#### 2.4.4. Characterization of Microscopic Composition

After the compression strength test, the samples were collected and soaked in absolute ethanol for 24 h to stop hydration. Appropriate bulk samples were selected for micromorphology analysis using scanning electron microscopy (SEM, The Hitachi SU 8010 field emission scanning electron microscope from Hitachi, Japan was used). These samples were plated by covering them with gold to make them conductive through the sputtering process in a duration of 90–150 s before the SEM test. The SEM analysis was carried out using a Japanese Hitachi SU8010 machine, the detection resolution, working distance and working voltage of which were fixed at 10 nm, 20–40 mm and 15 KV, respectively. Representative samples at certain hydration ages were crushed and ground using a pestle and mortar for XRD, FTIR and other tests. The composition of raw materials and the phases in hardened cement paste were checked using a D8-Discover X-ray diffractometer (XRD, The X-ray diffractometer used was a D8-Focus model from Bruker, Germany) with a CuKα1 radiation, a voltage of 40 kV, a current of 36 mA and a scanning speed of 3°/min. To analyze the molecular bonds and functional groups of the samples, FTIR testing was performed between the wavelengths of 500 to 4000 cm^−1^ on an infrared spectrometer (Thermo Fisher Scientific Nicolet IS50, Waltham, MA, USA) using the potassium bromide tablet pressing method. The hydration heat of the samples was analyzed using an isothermal calorimeter (TA/TAM AIR-8, TA Instruments, New Castle, DE, USA) at 25 °C for a duration of 72 h.

## 3. Results and Discussion

### 3.1. Enhancement of Pozzolanic Activity after Calcination

The chemical composition of the calcined attapulgite and montmorillonite was analyzed and the results are shown in the following Table 4. For the AG/CAG system, there were five main oxides, SiO_2_, Al_2_O_3_, MgO, Fe_2_O_3_ and CaO, in order of mass fraction, and the mass ratio of the main oxides did not change much after the high-temperature calcination treatment of CAG. For the MT/CMT system, similar to the AG/CAG system, the chemical composition did not change significantly after the high-temperature calcination treatment, and the major oxide components were SiO_2_, Al_2_O_3_, CaO, MgO and Fe_2_O_3_ in the order of mass fraction. First, the activity indices of the samples was measured on days 7, 14, 21 and 28, with the data of days 5, 18 and 25 interspersed to ensure the precision and reliability of the data.

Active SiO_2_ and Al_2_O_3_ in the calcined clay mineral powder samples reacted with Ca (OH)_2_ and the reaction products could be dissolved in hydrochloric acid solution. Additionally, CaO and other components in the powder samples could also be dissolved in hydrochloric acid solution. The active component content of sample reaction could be obtained by following this very principle, and the activity rate of the powder was obtained. In this experiment, active samples with different curing days were prepared to explore the reaction degree of powder with different curing periods. The change trend of the pozzolanic activity of these two calcined clays over time is shown in Figure 2.

As indicated by Figure 2a, the chemical reaction of CAG powder started after it was exposed to saturated lime water solution, and the amount of reaction increased gradually as time went by. The activity index of CAG powder was 15.76% at Day 7 and reached 47.98% at Day 21, and its growth rate tended to be stable at the later stage, reaching 54.22% at Day 28. As depicted in Figure 2b, similar to CAG, the chemical reaction of the CMT powder had begun to occur after contact with saturated lime water solution, and as time went by, the reaction amount increased gradually, achieving an activity index of 8.44% on Day 7 and 21.73% on Day 21; the growth rate tended to be gentle in the later period and reached 25.70% on Day 28.

It is noteworthy that the environment of the CAG/CMT in the saturated lime aqueous solution was different from that in the cementitious system. The former was in liquid solution, and its ion migration and other obstacles were relatively small, while in the case of the latter, because of the hydration reaction of cement clinker, the generation of hydration products would inevitably wrap and hinder the active reaction of the CAG/CMT powder. Therefore, the effect evaluation could laterally reflect the favorable pozzolanic activity of CAG/CMT, but failed to quantitatively describe the reaction process and amount of CAG/CMT in the cementitious system.

### 3.2. Influence of Calcination on Attapulgite and Montmorillonite

After being calcined at 750 °C, the particle-size microstructures of attapulgite and montmorillonite were significantly changed. As indicated by Figure 3, the median particle size of both attapulgite and montmorillonite after calcination was 7.404 μm larger than that of cement, and the median particle size of attapulgite after calcination was the largest, reaching 16.3 μm, which was speculated to result from the agglomeration of attapulgite or montmorillonite powder into large particle sizes after sintering.

Figure 4 depicts the microstructure changes of attapulgite and montmorillonite before and after calcination. As shown in Figure 4a, the microstructure of attapulgite before calcination presented a parallel, compacted and disordered arrangement of needle-like fibers, and there were micro-pores about 2 μm between each crystal. Crystal impurities similar to sheets could be clearly observed that were presumed to be the existence of hydromica (illite) and other impurities in the original sample. The microstructure of the calcined attapulgite was still like acicular fiber, but its parallel arrangement was broken, and the winding effect was more prominent. As shown in Figure 4b, the microscopic state of cloud-like sheet stacking was retained after the calcination of montmorillonite. Meanwhile, it could be clearly noticed that the size of the sheet structure was reduced after being calcined, and it was refined into more fine sheet structures stacked together. By analyzing the microstructures of attapulgite and montmorillonite after calcination, the above inference was verified that the particle sizes of attapulgite and montmorillonite increased after calcination.

Figure 5 describes the X-ray diffraction patterns of attapulgite and montmorillonite before and after calcination. In attapulgite (Figure 5a), the main crystal phase was attapulgite, involving a small amount of hydromica (illite) and quartz phase impurities. After calcination at 750 °C, the main diffraction peak corresponding to the attapulgite crystal phase disappeared and only the quartz phase was observed. In comparison, in montmorillonite (Figure 5b), the most important crystal phase was montmorillonite, containing a small number of impurities such as sodimica, albite, quartz and ettringite. After high temperature calcination, the main diffraction peak of the montmorillonite crystal phase disappeared, which also indicated that the crystal structure of montmorillonite was seriously damaged after high temperature, while the diffraction peak of the remaining impurities did not change significantly. Additionally, it was not accompanied by the generation of new crystalline phases, so it could be inferred that amorphous phase components might be generated in the process of calcination or conversion of the existing crystalline phase [23].

### 3.3. Effect of Working Performance and Mechanical Strength

#### 3.3.1. Fluidity

In concrete clay, mineral materials are mainly manifested in the reduction of water-reducing-agent dispersion, reducing the work performance of fresh concrete. Although concrete and cement slurry are very different, they are inextricably linked to each other. Gaining further knowledge about cement slurry is a good guide for concrete batching design and functional application.

The effects of different mixtures of attapulgite and montmorillonite on the working properties of cement paste before and after calcination are shown in Figure 6 below.

In the AG/CAG group, the clean cement paste system without CAG could be automatically amortized, and the fluidity reached 245 mm during the test. As the AG content increased to 1 wt%, the fluidity of the cementing material decreased sharply to 60 mm and remained stable then, which was attributed to the fact that AG would absorb a large amount of water-reducing agent in the mixing process so that it could not be well dispersed in the system or play the role of water reducing, and due to the unique pore structure of AG, it would further absorb the free water content of the slurry, thereby resulting in a significant reduction in the fluidity of mixed cement paste [24]. Similarly, when 1 wt% of CAG was added, the fluidity of the cement paste system would also drop sharply to 80 mm. However, differently from the above, the crystal structure of attapulgite was damaged and the specific surface area was reduced after AG calcining at high temperature, thus resulting in the weakened adsorption of the water-reducing agent and free water by the cement paste. In this case, mixing 2 wt% or even more CAG would increase the oar degree of powder under the same conditions. In summary, attapulgite had a rather pronounced effect on reducing the fluidity of polycarboxylic acid cement propeller, and its viscosity was closely related to the generated hydration products [25].

In contrast, in the MT/CMT group, although the addition of MT and CMT would have an impact on the fluidity of the cement paste, the addition of MT had a greater impact on the fluidity of cement paste. When 2 wt% was added, the fluidity of cement pastes almost completely disappeared and decreased to 60 mm, which might be attributed to the intercalation reaction between polycarboxylic acid water-reducing agent and aluminum silicate laminates. The microstructure of MT was lamellar, and polycarboxylic acid water-reducing agent had a high affinity with aluminum silicate layers in the MT structure, which seriously affected the further dispersion of water-reducing agent in the cementitious system, thus reducing its fluidity [26]. The addition of CMT would also reduce the fluidity of the original cementing material, but its reduction range was much smaller than that of MT. This was because the crystal structure of CMT had been destroyed after high-temperature calcination, and the adsorption effect was greatly reduced. The added water-reducing agent could be better dispersed in the cementing system compared to the MT group, resulting in a much smaller range of mobility reduction. The micro-mixing of the two could lead to a great loss in slurry flow, and calcination treatment could improve this effect from a mineralogical point of view, with a more obvious effect.

#### 3.3.2. Compressive Strength

The development of compressive strength of the cement paste samples with increase in age and different raw material contents is presented in Figure 7, where it can be seen that with increasing age, the compressive strength of the samples also kept rising, and the samples of various proportions all reached optimal strength at Day 28. However, with the age remaining unchanged, adding a small amount of CAG could indeed improve the compressive strength of the cement samples. It can also be observed from Figure 7 that Group A6 (with a CAG content of 6 wt%) had the best compressive strength enhancement effect and the content proportion was the best. At Day 3, Day 7 and Day 28, the compressive strength of A0 (with a CAG content of 0 wt%) in the control group was increased by 11.6%, 17.5% and 9.5%, respectively. However, the addition of a small amount of CMT in the early stage caused a slight decrease in the compressive strength of cement paste compared with the control group. This might be because the CMT in the early stage was not as reactive as the cement clinker and acted as an inert component in the reaction system, thus making the strength of the samples doped with CMT less than that of the control group. Then, 7 days later, the strength of the control group added with CMT was the same as that of the control group, which might be related to the reactivity mentioned above. At this time, the reactivity of CMT gradually increased until it was even with cement clinker. After 28 days, the strength of the control group doped with CMT became higher than that of the control group, which was also confirmed. It could be observed that in the case of a content of 8%, the strength value reached its maximum, i.e., about 85 MPa. In addition, the addition of CMT was beneficial to the improvement of the mechanical properties of the sample in the later stage.

### 3.4. Microstructure

Figure 8 compares the micromorphologies of the control group and the calcined attapulgite system mixed with 8 wt% at different ages and presents the micromorphologies after 3, 7 and 28 days under standard curing, respectively. As curing time went by, both the blank group and the control group demonstrated a process of evolution from loose granular structure to dense network structure. Since the dosage of CAG was small and no new elements were introduced in the process of CAG mixing, the hydration process of the gel system was not significantly affected, and was still consistent with the traditional OPC hydration process. At Day 3, FO in the control group exhibited clear acicular hydration products, and these disordered acicular structures were mainly calcium silicate hydrate (C-S-H) hydration products and a small amount of ettringite hydration products, while CA8 group hydration products demonstrated more network structures and fine acicular structures. The introduction of CAG significantly improved the microstructure and properties of the hydration products [27]. At Day 7, the hexagonal morphology of hydrated calcium hydroxide (CH) could be clearly observed in the control group FO, and the previous disordered needle structure disappeared substantially. In addition, the hardening and tending to the dense structure of hydration products appeared. However, no obvious hexagonal CH was found in the CA8 group, the hardening phenomenon was more obvious and the sampled parts tended to be a whole structure. This was because the addition of CAG chemically reacted with the CH in the cement hydration products to produce hydration products such as C-S-H. In addition, the flocs produced by clay were highly stable flocs with a strong interaction effect between particles preventing the destruction of the flocs [28]. After 28 days of curing, the tricalcium silicate in the cement clinker was greatly involved in the hydration process, and both groups exhibited an obvious compact structure. Interestingly, the former had a partial directional trend in the local hydration products causing cracks, while the latter was better combined into a compact overall structure. Undeniably, the fibrous structure of CAG itself was also conducive to the further development of the hydration product network structure [29].

Figure 9 compares the micromorphologies of the control group and the calcined montmorillonite system mixed with 10 wt% at different ages and presents the micromorphologies after 3, 7 and 28 days under standard curing. It can be preliminarily observed by comparing the microstructure that the microstructure of both the control group (FO) and the experimental group (CM10) developed from loose granular structure to dense network structure as the curing time went by. In terms of the overall content, the content of CMT was small and no new elements would be introduced to the mixing process, so the hydration process could be roughly regarded as consistent with cement hydration and still in line with the traditional OPC hydration process.

From the perspective of longitudinal comparison, the control group (FO) had a loose needle-like structure from early curing at Day 3, and the hydration products of these needle-like disordered structures mainly contained calcium carbonate hydrate and a small amount of ettringite hydrate products, while the hydration products at Day 7 could be obviously observed as hexagonal calcium hydroxide (CH) hydrate. In addition, a large number of loose needle-like structures disappeared, the hydration products tended to plate and the structure was denser than that at Day 3. As the curing period was prolonged to 28 days, the tricalcium silicate in the cement clinker was heavily involved in the hydration process, presenting an obvious plate dense structure. It was noteworthy that obvious cracks could be observed at Day 28, and the directional trend of hydration products was speculated to be the cause of the generation of cracks. From the perspective of cross-sectional comparison, the trend of all microstructures was towards the development of densification, with the control group (F0) showing clear needle-like hydration products (C-S-H gels and a small amount of ettringite) from early maintenance at day 3, and the hydration products in the CM10 group showing more of a network-like structure prototype, indicating that the introduction of CMT was able to improve the microfine organization of the early hydration products and properties [27]. At day 7, the hexagonal shape of hydrated calcium hydroxide (CH) was clearly visible in the control group (F0) and tended to be a dense structure, while the hexagonal shape of hydrated calcium hydroxide (CH) was not obvious in the microstructure of the CM10 group, the hydration products were more refined and the slabbing phenomenon was more obvious and tended to be a whole. This was because CMT can chemically react with CH in the hydration products of cement to produce hydration products such as C-S-H. In addition, the flocs produced by clay were highly stable flocs with a strong interparticle action effect that could prevent the flocculation products from being destroyed [28]. At day 28, both the control (F0) and CM10 groups exhibited a clear slab dense structure, but the former exhibited cracks and the latter was more viscous and had better overall properties. This is because the tricalcium silicate in the cement clinker had been substantially involved in the hydration process. Overall, the results of the micromorphology were consistent with the previous tests of mechanical properties.

### 3.5. Phase Composition Analysis

During the exposure of the cement clinker to water, firstly, tricalcium aluminate (C_3_A) reacted chemically with gypsum and other components to produce ettringite (AFt), which provides early strength for cementitious materials; this is the initial hydration reaction of cement. It is worth mentioning that when the entire system is depleted of gypsum, ettringite (AFt) is converted to monosulfide hydrated calcium sulfaluminate (AFm). Subsequently, dicalcium silicate (C_2_S) as well as tricalcium silicate (C_3_S) react chemically with water to produce hydration products such as calcium hydroxide (CH) and C-S-H gels, which provide the later strength of the whole system and are the main source of strength of cementitious materials. The diffraction peaks of hydration products such as CH and C-S-H gels became more and more obvious in our XRD patterns, while the diffraction peaks of components such as C_3_S in the cement clinker gradually weakened.

Figure 10 below depicts the XRD patterns of calcined attapulgite paste with different dosages at different ages. It can be observed from Figure 10a that as the curing time went by, the diffraction peaks of hydration products such as CH in the whole system became increasingly obvious, while those of cement clinker such as C_3_S gradually weakened, indicating a decrease in corresponding raw materials and an increase in the corresponding products of the hydration reaction. Figure 10b–d describe the XRD patterns of different dosages of CAG at 3 d, 7 d and 28 d, respectively. We found that the addition of CAG did not bring new chemical components to the whole system, and the types of hydration products did not change significantly. However, the addition of CAG provided additional chemical active components such as SiO_2_ and Al_2_O_3_ that would generate additional C-S-H products in a strong alkali environment, thereby facilitating the hydration process of proceeding in a positive direction and contributing to the development of strength [30].

Similar to CAG, the addition of CMT did not introduce new chemical components into the system during the whole hydration process. Similarly, in the hydration process, in terms of the strength of the diffraction peak, there was also a decrease in raw materials in cement clinker and an increase in hydration products, as shown in Figure 11, indicating that the hydration process was in a forward direction. Figure 11b–d show the XRD patterns of CMT with different dosages under different curing ages. It can be seen that the addition of CMT led to results consistent with those of the above CAG system, which are not repeated here.

### 3.6. FT-IR Analysis

Infrared analysis can provide more information about the hydration products in a cementitious system. The infrared spectra of the samples at different ages are shown in Figure 12, with Figure 12a added to the CAG group and Figure 12b added to the CMT group. As indicated by the figure, the absorption peak of C_2_S was around 999 cm^−1^ while that of sulfoaluminate was around 1110 cm^−1^. It can also be observed that the absorption peak of the pure cement sample group (A0) at this point gradually disappeared with the progress of the hydration reaction, which was consistent with the theory of the process of cement hydration. In the early hydration process, the main hydration product was ettringite (AFt), and the absorption peak corresponding to -OH in CH was about 3640 cm^−1^, while those corresponding to H_2_O were 3420 cm^−1^ and 1643 cm^−1^. In addition, ettringite corresponded to the asymmetric stretching vibration of SO_4_^2-^ at the absorption peak around 1110 cm^−1^, which, due to the corresponding AFt conversion to AFm in the later hydration process, disappeared gradually with the progress of the hydration reaction [31]. The absorption peaks at 900–1000 cm^−1^ and 874 cm^−1^ corresponded to Si-O in silicate, while the absorption peak at 1424 cm^−1^ corresponded to carbonate, which might be caused by the inevitable carbonization reaction during sample preparation or hydration.

It should be noted that as shown in Figure 12a, the introduction of CAG did not exert a huge impact on the types of hydration products. With the addition of CAG dosage and the increase in curing age, the main obvious change was the gradual disappearance of the absorption peak around 1110 cm^−1^, indicating that the introduction of CAG contributed to the formation of AFt in the early stage and its transformation in the later hydration process. Additionally, the absorption peak around 900–1000 cm^−1^ tended to migrate to the higher wave range with the increase in the hydration reaction time and the CAG dosage, suggesting that the Si-O bond gradually polymerized into a Si-O-Si bond [32]. Similarly, the introduction of CMT did not have much effect on the types of hydration products. As shown in Figure 12b, with the increase in CMT content and the increase in curing time, the main obvious change was the gradual disappearance of the absorption peak near 1110 cm^−1^, indicating that the introduction of CMT was conducive to the formation of AFt in the early stage and conversion in the later hydration process. Additionally, absorption peaks around 900–1000 cm^−1^ tended to migrate to higher wave segments with the increase in the hydration reaction time and the CMT content, which also indicated that Si-O bonds polymerized gradually to Si-O-Si bonds, and the degree of silico-oxygen tetrahedron polymerization in C-S-H increased. In addition, there were similar phenomena in the IR spectra of CAG-OPC, and the transformation was speculated to be related to pH reduction in the system [32].

### 3.7. Hydration Heat of Cement Pastes

The hydration heat was measured at room temperature, as shown in Figure 13 below. It was generally believed that the early hydration heat release could be divided into four or five stages, the acceleration and deceleration stages of which were the research focus. As shown in Figure 13, the release stage of the hydration heat of the calcined attapulgite and montmorillonite cement system was roughly the same as that of the traditional OPC hydration heat release stage. With the increase in CAG/CMT content, the hydration peak in the acceleration period was advanced, and the peak value was lower than that of the control group(A0), indicating that the incorporation of CAG/CMT accelerated the hydration rate of the cementitious system, since hydration products could be additionally generated on the surface of CAG/CMT, and CAG/CMT provided more active SiO_2_, Al_2_O_3_ and other components in the whole cementitious system to participate in the hydration reaction, thus accelerating the reaction, similar to the study by Kaminskas et al. [5]. However, the difference was that they concluded that the calcination treatment of montmorillonite had little effect on its pozzolanic activity. That being said, the equivalent substitution of CAG/CMT reduces the amount of clinker in a cementitious system, so the peak value would indeed be reduced. In the CAG/CMT-OPC systems, the heat flow was significantly higher than the control group (A0) after about 40 h of hydration, which indicates that the chemical reaction effect of CAG/CMT was better afterwards; the addition of CAG/CMT acted as a filler in the system and provided additional shear, which made the water–cement ratio of the cement clinker increase and the hydration space larger, this being favorable for the precipitation of hydration products [33]. Figure 13 presents the cumulative heat release of the whole hydration process. In the first few hours, all samples had high heat release, but after 16 h, the cumulative heat release of A0 in the control group was the highest, which is of great significance to avoid the cracking caused by early temperature increases in concrete structures [34].

## 4. Conclusions

Herein, the pozzolanic activity of attapulgite and montmorillonite was enhanced using calcination, and a series of mixtures was prepared by replacing OPC with calcined attapulgite and montmorillonite. The hydration products and basic properties of these mixtures were correspondingly studied, and the following conclusions were drawn based on the results and analysis:

(1) After calcination, the pozzolanic activities of CAG and CMT were significantly enhanced with time, and the addition of CAG and CMT seriously reduced the fluidity of the mixed cement paste, achieving a maximum reduction of 75.5%. Additionally, it was found through testing the mechanical strength of the samples that adding a trace amount of calcined raw materials increased their compressive strength in the late hydration stage;

(2) Both the introduction of CAG and CMT contributed to the formation of AFt in the early stage and conversion in the later hydration process, which promoted the gradual polymerization of Si-O bonds to Si-O-Si bonds and increased the degree of silico-oxygen tetrahedron polymerization in C-S-H. The difference between the two was that CAG highlighted the peak generated by secondary ettringite, while CAG did not have a peak generated by secondary ettringite;

(3) Given that additional hydration products were generated on the surface of CAG/CMT, hydration in the acceleration period was advanced with the increase in content of the calcined raw materials. However, simultaneously, after the addition of the calcined raw materials, since there was more replacement of the original cement clinker, the cumulative heat release was reduced, which is of great significance to limit the early thermal cracking of concrete buildings.

## Figures and Tables

**Figure 1 materials-16-01794-f001:**
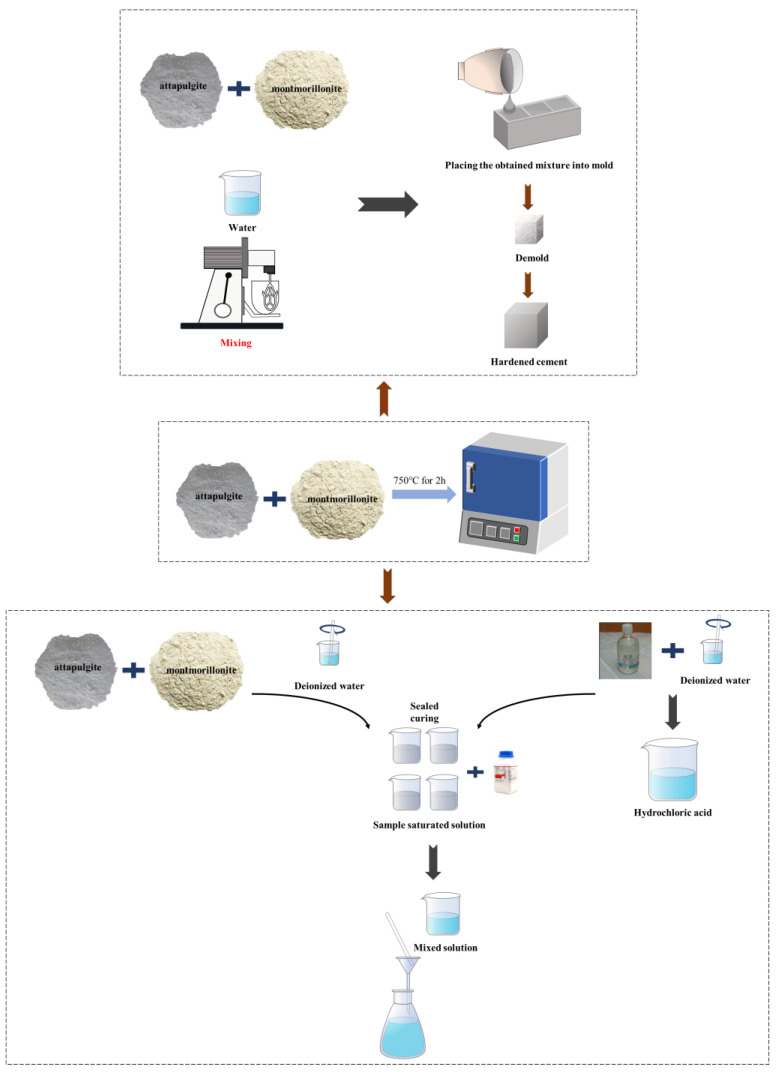
The flow chart.

**Figure 2 materials-16-01794-f002:**
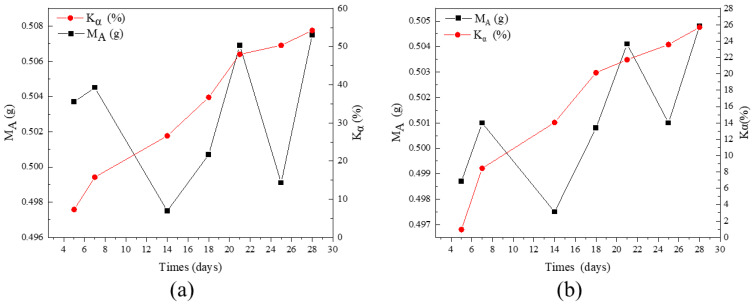
Determination of pozzolanic activity using the residue method: (**a**) CAG; (**b**) CMT.

**Figure 3 materials-16-01794-f003:**
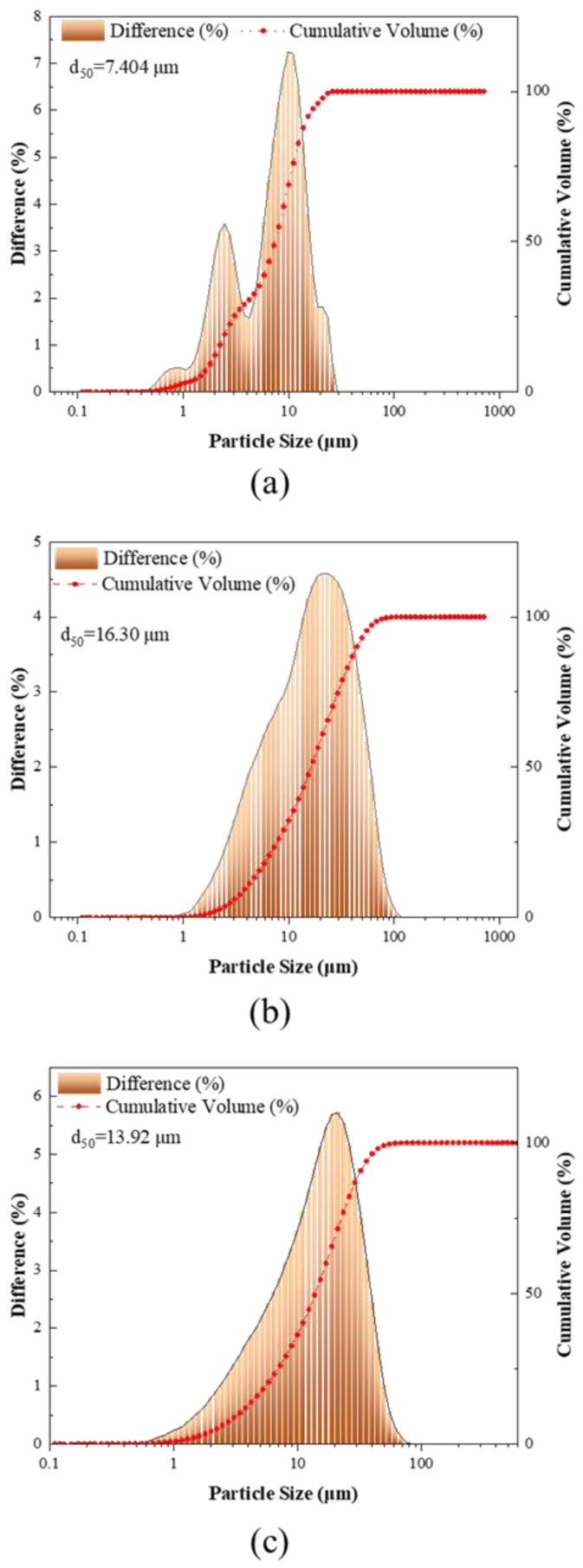
Particle size distribution: (**a**) OPC; (**b**) CAG; (**c**) CMT.

**Figure 4 materials-16-01794-f004:**
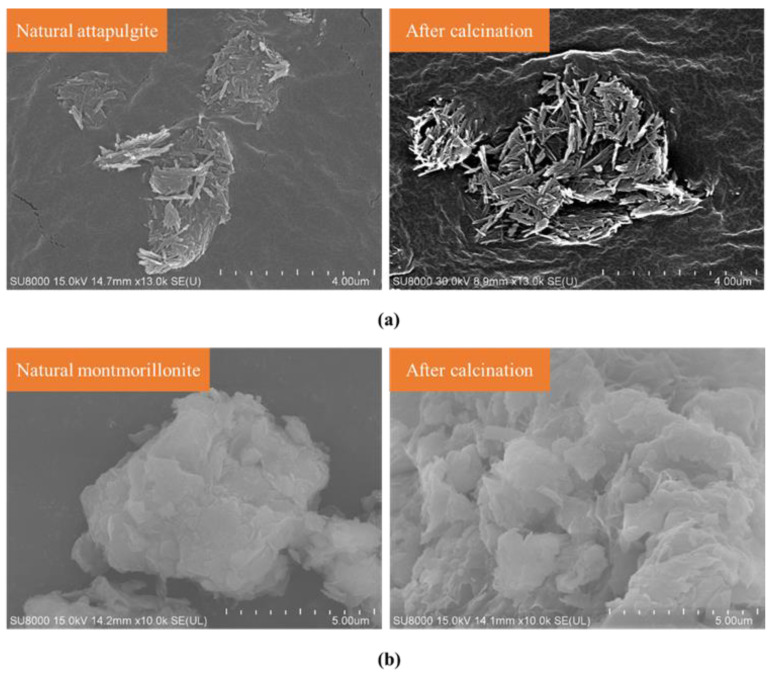
Microtopography: (**a**) attapulgite; (**b**) montmorillonite.

**Figure 5 materials-16-01794-f005:**
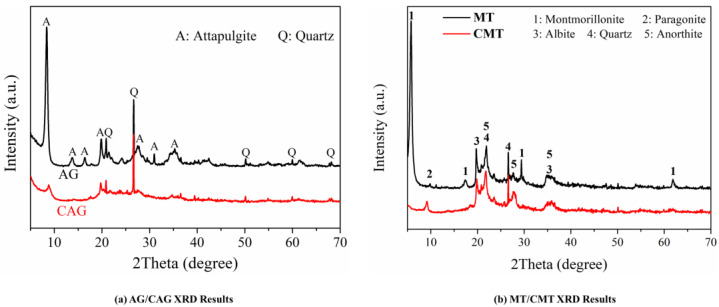
Phase composition analysis of clay before and after calcination: (**a**) attapulgite; (**b**) montmorillonite.

**Figure 6 materials-16-01794-f006:**
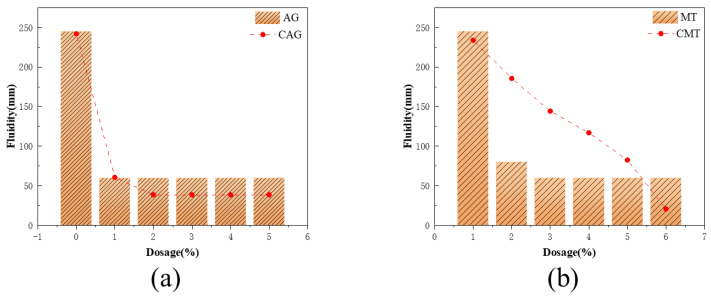
Effects of different contents of attapulgite and montmorillonite on the fluidity of cement paste before and after calcination: (**a**) attapulgite; (**b**) montmorillonite.

**Figure 7 materials-16-01794-f007:**
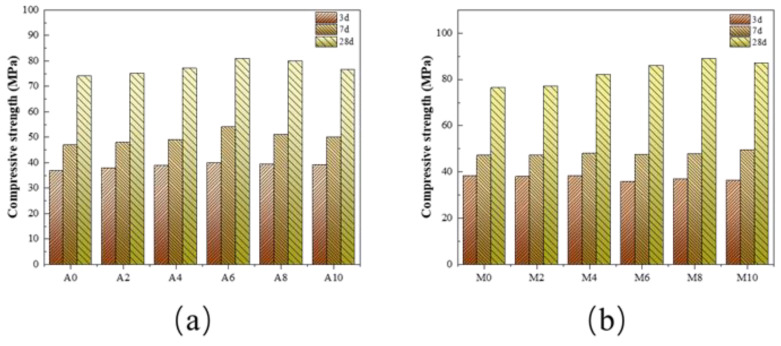
Compressive strength of cement paste samples: (**a**) attapulgite cement paste before and after calcination; (**b**) montmorillonite cement paste before and after calcination.

**Figure 8 materials-16-01794-f008:**
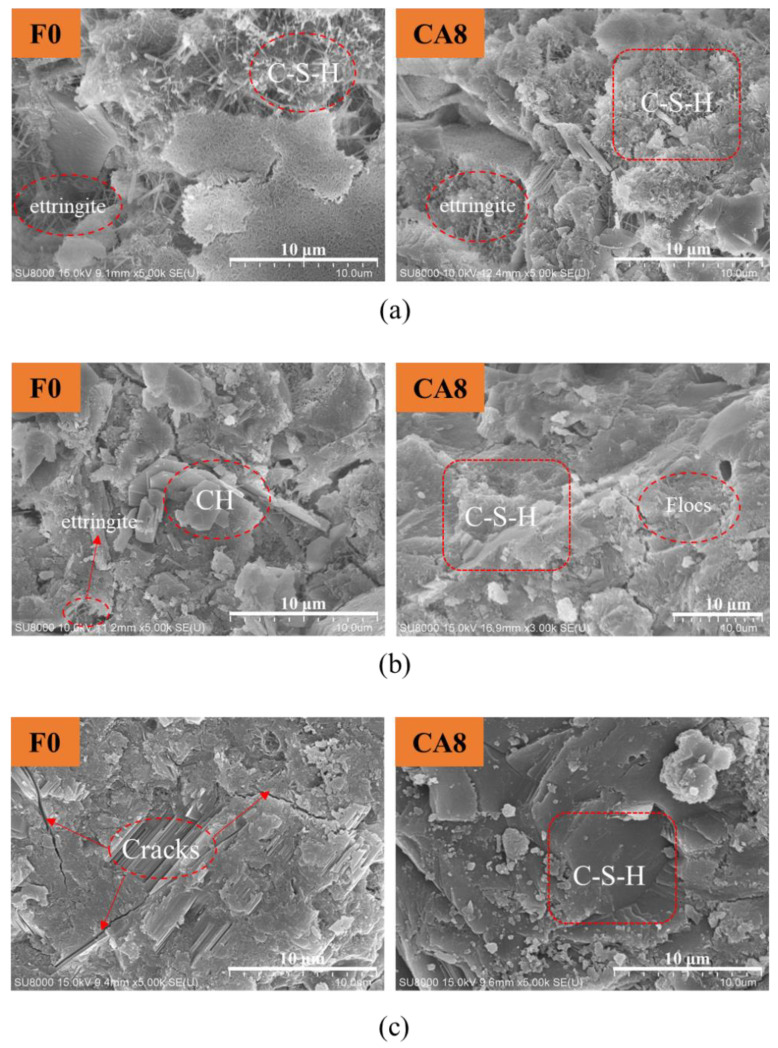
Micromorphology of cement paste at different ages before and after adding calcined attapulgite: (**a**) 3 d; (**b**) 7 d; (**c**) 28 d.

**Figure 9 materials-16-01794-f009:**
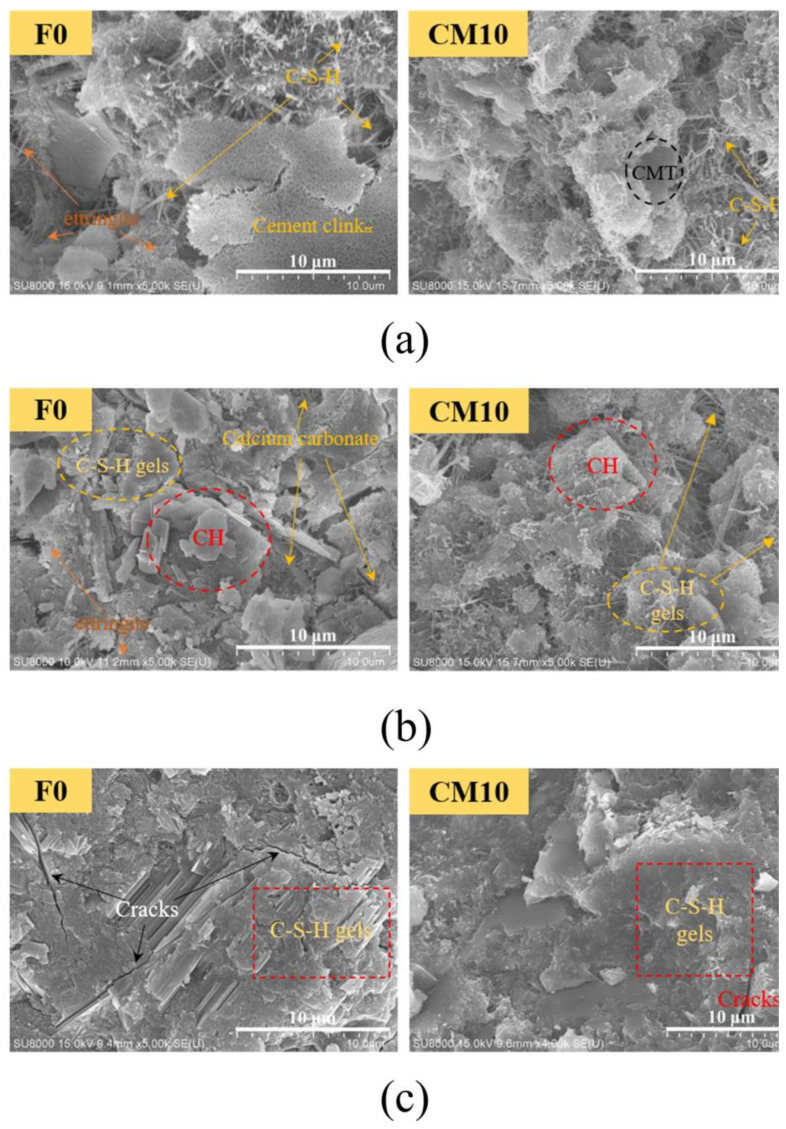
Micromorphology of cement paste at different ages before and after adding calcined montmorillonite: (**a**) 3 d; (**b**) 7 d; (**c**) 28 d.

**Figure 10 materials-16-01794-f010:**
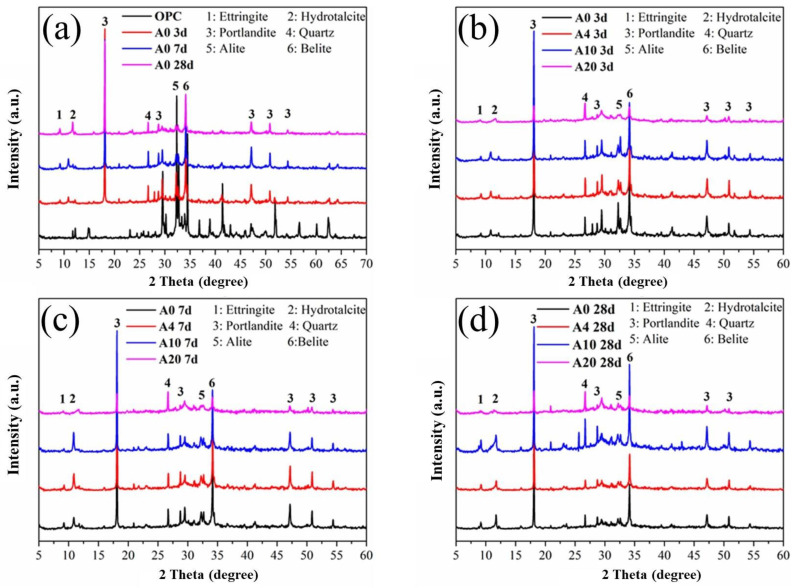
Phase composition analysis of calcined attapulgite cement slurry mixed with different amounts at different ages. (**a**) Control group(A0) at different age; (**b**) Different dosage at day 3; (**c**) Different dosage at day 7; (**d**) Different dosage at day 28.

**Figure 11 materials-16-01794-f011:**
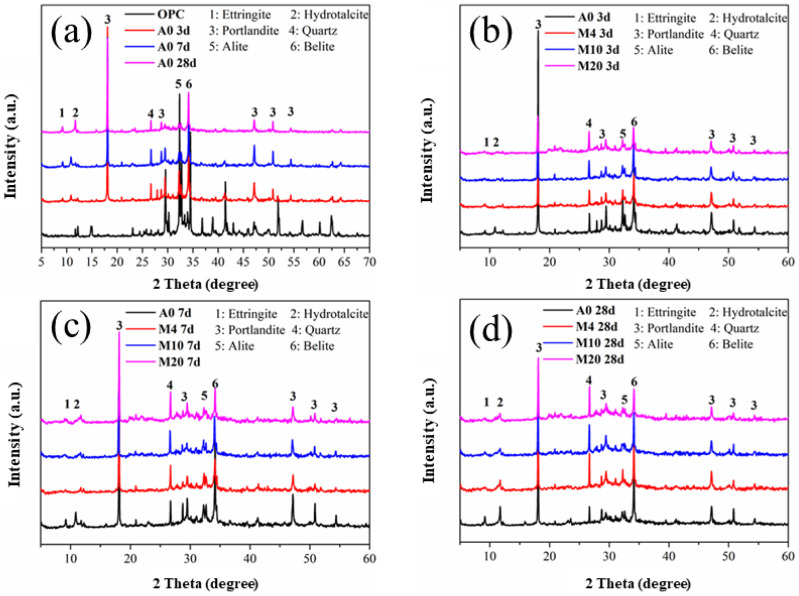
Phase composition analysis of calcined montmorillonite cement paste mixed with different amounts at different ages. (**a**) Control group(A0) at different age; (**b**) Different dosage at day 3; (**c**) Different dosage at day 7; (**d**) Different dosage at day 28.

**Figure 12 materials-16-01794-f012:**
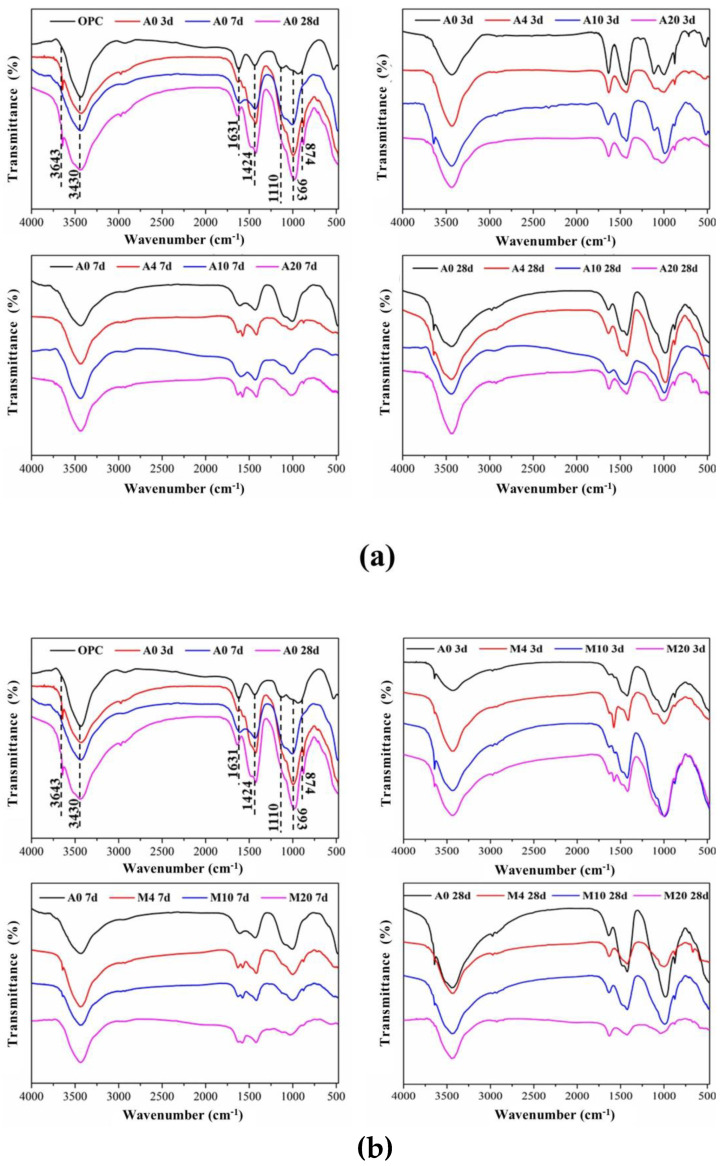
Infrared spectra of samples of different proportions at different ages: (**a**) attapulgite; (**b**) montmorillonite.

**Figure 13 materials-16-01794-f013:**
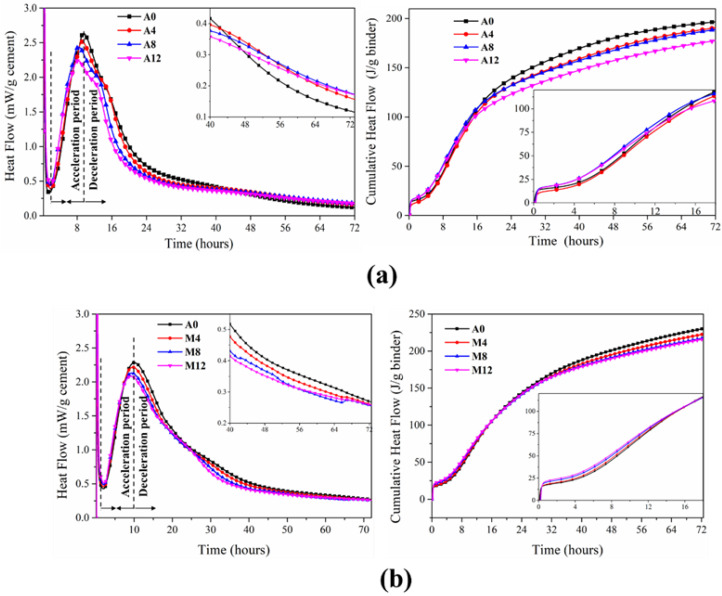
Heat release of cement pastes with time change: (**a**) CAG; (**b**) CMT.

**Table 1 materials-16-01794-t001:** Basic mechanical properties of Portland cement.

Specific Surface Area/m^2^·kg^−1^	Setting Time/min		Flexural Strength/MPa		Compressive Strength/MPa	
336	Initial	Final	3 d	28 d	3 d	28 d
	190	280	6.4	11.2	23.2	44.5

**Table 2 materials-16-01794-t002:** Main chemical compositions of Portland cement (OPC), attapulgite (AG) and montmorillonite (MT).

Material	Mass Fraction/%								
	Na_2_O	MgO	Al_2_O_3_	SiO_2_	TiO_2_	K_2_O	CaO	Fe_2_O_3_	LOI
Cement	0.14	1.72	5.94	22.58	3.60	/	0.71	59.20	2.27
AG	0.07	11.26	11.70	69.22	0.01	/	0.96	1.24	5.37
MT	0.25	2.24	15.34	75.92	/	0.16	1.2	3.18	1.14

**Table 3 materials-16-01794-t003:** Proportion of cement slurry sample preparation.

Serial Number	Label	Cement/g	AG(CAG)/g	MT(CMT)/g	Water Reducing Agent/g	Water/g
1	F0	400	0	/	0.4	116
2	A1	396	4	/	0.4	116
3	A2	392	8	/	0.4	116
4	A3	388	12	/	0.4	116
5	A4	384	16	/	0.4	116
6	A5	380	20	/	0.4	116
7	CA1	396	4	/	0.4	116
8	CA2	392	8	/	0.4	116
9	CA3	388	12	/	0.4	116
10	CA4	384	16	/	0.4	116
11	CA5	380	20	/	0.4	116
12	M1	396	/	4	0.4	116
13	M2	392	/	8	0.4	116
14	M3	388	/	12	0.4	116
15	M4	384	/	16	0.4	116
16	M5	380	/	20	0.4	116
17	CM1	396	/	4	0.4	116
18	CM2	392	/	8	0.4	116
19	CM3	388	/	12	0.4	116
20	CM4	384	/	16	0.4	116
21	CM5	380	/	20	0.4	116

**Table 4 materials-16-01794-t004:** Chemical fractions of attapulgite and montmorillonite before and after calcination (mass %).

Oxide	Na_2_O	MgO	Al_2_O_3_	SiO_2_	SO_3_	K_2_O	CaO	Fe_2_O_3_	TiO_2_
AG	0.07	11.26	11.70	69.22	0.01	0.96	1.24	5.37	0
CAG	0.34	10.20	12.71	69.34	0.02	0.89	1.81	4.60	0
MT	0.25	2.24	15.34	75.92	0	1.2	3.18	1.14	0.16
CMT	0.28	2.2	15.42	76.21	0	1.21	3.21	1.12	0.16

## Data Availability

Not applicable.

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
