# Peer review of "Exploration of the Compressive Strength and Microscopic Properties of Portland Cement Taking Attapulgite and Montmorillonite Clay as an Additive"

_materials, 2023, doi:10.3390/ma16051794_

Round 1

Reviewer 1 Report

Manuscript ID: materials-2191353

Manuscript Title: Exploration on the compressive strength and microscopic property of Portland cement taking attapulgite and montmorillonite clay as an additive

The following corrections are recommended:

1.      Please rewrite the statement "The effects of attapulgite and montmorillonite on the working properties, mechanical strength, phase composition, morphology, hydration and heat release of Ordinary Portland Cement (OPC) were studied with the cement calcinated at 750℃ for 2 h as Supplementary Cementing Materials (SCMs)" as it is not well understood.

2.      The novelty of the study and its difference from the literature was not mentioned.

3.      There is a lack of review of research on montmorillonite.

4.      It was better to examine the chemical composition after calcination and discuss its effect on the oxide's ratios (before and after).

5.      Why is the calcining temperature of 750°C specifically adopted, please explain.

6.      The authors mention the following statement on page 3 "Herein, the residual quantity method summarized by Zhong", but the claim is not cited by a reference.

7.      Most of the interpretations and discussions in the Results and Discussion section have not been supported by a reference (s) nor have they been compared to the literature. So, this section requires major improvement.

8.      More elaborations should be added to the SEM images in Figures 8 and 9.

9.      Conclusions should be improved by shortening them and focusing on the main findings only.

10.  The bibliography should be reinforced by other recent references.

Reviewer 2 Report

An interesting topic of research about the effects of attapulgite and montmorillonite on the working properties, mechanical strength, phase composition, morphology, hydration and heat release of Ordinary Portland Cement (OPC). Authors should highlight the novelty of the work.

In the current study, two types of clays were used as additive components for the improvement of OPC, the introduction could be extended with more information about other additives including natural clay that was previously published such as "Some applications of clays in radioactive waste management." Clays and clay minerals: geological origin, mechanical properties and industrial applications. Hauppauge, NY, USA: Nova Science Pub. Inc., 2014. 403-415.

I find the manuscript is well structured however, I recommend more clearly emphasizing the novelty of this study in the introduction and conclusion, in order to show the contribution of the authors in this field obtained in this study.

With the mechanical and morphological investigations carried out in the study, the porosity and density will be more important to explain the effect on compressive strength.

Moreover, in parallel with the heat of hydration, thermal conductivity is required to be evaluated.

The discussion needs more deep investigation.

The conclusion could include a recommendation about the proper applications of the nominated admixtures.

The format of the reference citation needs to be maintained. So please check and revise accordingly to the journal's instructions.

In conclusion, I believe that the theme of this manuscript can be consistent with the theme of materials. At the same time, the manuscript needs to be improved. Authors should edit the manuscript in accordance with the guidelines mentioned above.

Reviewer 3 Report

The proposed work focuses on the exploration on the compressive strength and microscopic property of Portland cement taking attapulgite and montmorillonite clay as an additive. It is of potential interest to Materials journal readers.

Despite the importance of the subject addressed, this work needs many improvements to be ready for the publication in the Materials journal. 

Specific points of improvement :

-        Literature review section is too weak. So, it must be improved by more previous researches.

-        The objective of this research must be more developed by a comparison with previous researches results.

-        The section "Experiment" should be replaced by a "Materials and methods". In this section, all test standards must be indicated.

-        Explain why The highest pozzolanic activity index of attapulgite was 54.22% at Day 28, while that of montmorillonite was 25.7% at Day 28.

-        Results need an in-depth discussion.

-        Quality of figures must be improved.

-        The conclusion section is too long. Only the main results must be indicated.

Round 2

Reviewer 1 Report

The authors have improved their manuscripts. No additional modifications are required.

Reviewer 2 Report

Accepted for publication in the current from.

Reviewer 3 Report

The proposed work focuses on the exploration on the compressive strength and microscopic property of Portland cement taking attapulgite and montmorillonite clay as an additive. It is of potential interest to Materials journal readers.

I think that the revised version of the submitted paper is well improved by considering the reviewers and editor recommendations and remarks. Indeed, I think that this paper is accepted in this form.